# Exploring the Limit of Outcome Reward for Learning Mathematical Reasoning

**Chengqi Lyu**[1*]    **Songyang Gao**[1*]    **Yuzhe Gu**[1,2*]    **Wenwei Zhang**[1*†]    **Jianfei Gao**[1]
**Kuikun Liu**[1]    **Ziyi Wang**[1]    **Shuaibin Li**[1]    **Qian Zhao**[1]    **Haian Huang**[1]
**Weihan Cao**[1]    **Jiangning Liu**[1]    **Hongwei Liu**[1]    **Junnan Liu**[1]    **Songyang Zhang**[1]
**Dahua Lin**[1,3,4]    **Kai Chen**[1†]
[1]Shanghai AI Laboratory    [2]Shanghai Jiao Tong University
[3]MMLab, The Chinese University of Hong Kong    [4]HKGAI under InnoHK
{lvchengqi,gaosongyang,guyuzhe,zhangwenwei,chenkai}@pjlab.org.cn

## Abstract

Reasoning abilities, especially those for solving complex math problems, are crucial components of general intelligence. Recent advances by proprietary companies, such as o-series models of OpenAI, have made remarkable progress on reasoning tasks. However, the complete technical details remain unrevealed, and the techniques that are believed certainly to be adopted are only reinforcement learning (RL) and the long chain of thoughts. This paper proposes a new RL framework, termed OREAL, to pursue the performance limit that can be achieved through **O**utcome **RE**w**A**rd-based reinforcement **L**earning for mathematical reasoning tasks, where only binary outcome rewards are easily accessible. We theoretically prove that behavior cloning on positive trajectories from best-of-N (BoN) sampling is sufficient to learn the KL-regularized optimal policy in binary feedback environments. This formulation further implies that the rewards of negative samples should be reshaped to ensure the gradient consistency between positive and negative samples. To alleviate the long-existing difficulties brought by sparse rewards in RL, which are even exacerbated by the partial correctness of the long chain of thought for reasoning tasks, we further apply a token-level reward model to sample important tokens in reasoning trajectories for learning. With OREAL, for the first time, a 7B model can obtain 94.0 pass@1 accuracy on MATH-500 through RL, being on par with 32B models. OREAL-32B also surpasses previous 32B models trained by distillation with 95.0 pass@1 accuracy on MATH-500. Our investigation also indicates the importance of initial policy models and training queries for RL. Code, models, and data are available at https://github.com/InternLM/OREAL.

## 1 Introduction

Solving complex problems with reasoning capability forms one of the cornerstones of human cognition - a cognitive ability that artificial general intelligence must ultimately master (Xu et al., 2025; Zhong et al., 2024). Among various problem domains, the mathematical problem emerges as a particularly compelling experimental paradigm for AI research (Liu et al., 2023; Matzakos et al., 2023; Ying et al., 2024; Shao et al., 2024), owing to its relatively well-defined structure and availability of precise binary correctness feedback based on the verifiable final answers.

Recent advances in large language models (LLMs) have achieved remarkable progress in mathematical reasoning by the chain-of-thought technics (Wei et al., 2022; Wang et al., 2022;

---

*Equal contribution † Corresponding author

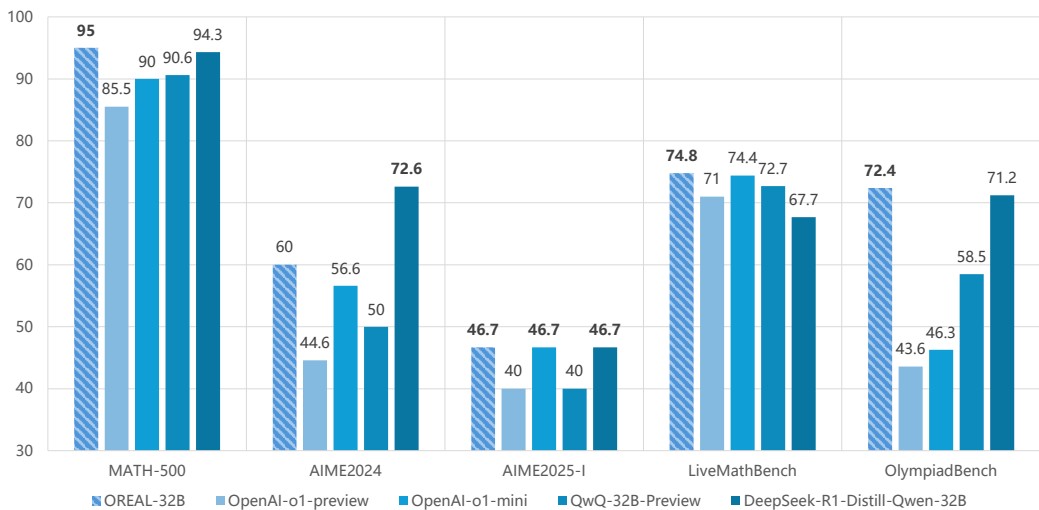

Figure 1: Overall performance between OREAL-32B and some competitive baselines.

Kojima et al., 2022), in which the LLMs are elicited to produce a series of intermediate reasoning steps before providing the final answers to the problem. However, as most of the capable models (*e.g.,* the o-series models by OpenAI (OpenAI, 2024b)) are developed by proprietary companies, there is no clear pathway to develop state-of-the-art reasoning models. Some recent work shows that distillation (Huang et al., 2024; Min et al., 2024) is sufficient to obtain high performance given the accessibility to existing best or near best AI models, reinforcement learning (RL) is believed to be a more fundamental approach and has exhibited potential (DeepSeek-AI et al., 2025) to advance beyond the intelligence boundary of current AI models, using the most capable open-source foundation models (DeepSeek-V3-base (Liu et al., 2024a), *inter alia*).

However, fundamental challenges of sparse reward in RL persist and are even exacerbated in mathematical reasoning tasks that mainly rely on the chain of thought technics with LLMs (Wei et al., 2022): the evaluation of intermediate reasoning steps is labor intensive (Lightman et al., 2023) and its accurate automation approach is still under-explored, thus, the only reliable reward is based on the outcome (correctness of final answer), which is inherently binary and sparse when faced with more than 2000 tokens in the long reasoning trajectories (DeepSeek-AI et al., 2025; Team et al., 2025). Existing approaches have attempted to estimate the advantages or values of reasoning steps by search (Wang et al., 2024; Kazemnejad et al., 2024) or value function-based credit assignment (Schulman et al., 2017; Cui et al., 2025), yet, their performance remains unsatisfactory in comparison with the distilled models (DeepSeek-AI et al., 2025).

This paper aims to conquer the above challenges and proposes a simple framework, termed OREAL, to push the limit of **O**utcome **REwA**rd-based reinforcement **L**earning for mathematical reasoning tasks. OREAL is grounded in the unique characteristics of mathematical reasoning tasks that binary outcome feedback creates an environment where all positive trajectories are equally valid. We first establish that behavior cloning on BoN-sampled positive trajectories is sufficient to achieve KL-regularized optimality, which emerges from the analysis that the positive trajectory from BoN sampling converges to a distribution independent of the sample number. For learning on negative samples, OREAL reveals the necessity of reward shaping to maintain consistent gradient estimation between sampling and target distributions. Such a mechanism compensates for BoN's under-sampling of negative gradients, and enables difficulty-adaptive optimization over both successful and failed trajectories.

Another intrinsic property of mathematical reasoning tasks is the partial correctness in long reasoning chains, which further imposes the learning difficulty of sparse rewards when only a binary outcome reward is available at each iteration of RL training. Thus, OREAL adopts

a lightweight credit assignment scheme through a token-level reward model trained using outcome rewards. This mechanism automatically estimates step-wise importance weights by decomposing trajectory advantages, enabling focused learning of critical reasoning steps or errors. The integration of these components yields a theoretically sound framework that effectively bridges the gap between sparse binary feedback and dense policy optimization requirements for mathematical reasoning tasks.

Extensive experimental results show that OREAL effectively improves the mathematical reasoning capability of LLMs. At the 7B parameter scale, to the best of our knowledge, OREAL-7B is the first to obtain the pass@1 accuracy on MATH-500 (Hendrycks et al., 2021) to 91.0 using RL instead of distillation, which even exceeds QwQ-32B-Preview (Team, 2024) and o1-mini (OpenAI, 2024b). OREAL also improves DeepSeek-R1-Distilled-Qwen-7B from 92.8 to 94.0 pass@1 accuracy, being on par with the previous best 32B models. For the 32B model, OREAL-32B outperforms all previous models (Figure 1), both distilled and RL-based, obtaining new state-of-the-art results with 95.0 pass@1 accuracy on MATH-500.

## 2 Methods

In this section, we first analyze the formulation of RL and the intrinsic properties of underlying binary feedback environments (§2.1), and establish a theoretical foundation for our optimization framework about how to learn from long reasoning chains.

### 2.1 Preliminary

When adopting a large language model (LLM) for mathematic reasoning, common practices (Shao et al., 2024; Yuan et al., 2023) conduct binary feedback (0/1 reward) based solely on the correctness of LLM's final answer, and perform policy optimization correspondingly.

**Policy Optimization** Considering a Markov Decision Process (MDP) defined by the tuple $(\mathcal{S}, \mathcal{A}, P, r, \gamma)$, where $\mathcal{S}$ is a finite state space, $\mathcal{A}$ is the action space, $P(s'|s, a)$ specifies the state transition dynamics, $r : \mathcal{S} \times \mathcal{A} \to \mathbb{R}$ is the reward function, and $\gamma \in [0, 1)$ denotes the discount factor. We focus on KL-regularized policy optimization, which can be formulated as:

$$J(\theta) \triangleq \mathbb{E}_{s \sim \rho_0, a \sim \pi_\theta(\cdot|s)} \left[ Q^{\pi_\theta}(s, a) \right] - \alpha \cdot \mathbb{E}_{s \sim \rho_0} \left[ D_{\mathrm{KL}} \left( \pi_\theta(\cdot|s) \| \pi_0(\cdot|s) \right) \right] \tag{1}$$

with the state-action value function $Q^\pi(s, a) \triangleq \mathbb{E}_\pi \left[ \sum_{k=0}^\infty \gamma^k r(s_{t+k}, a_{t+k}) \mid s_t = s, a_t = a \right]$ under vanilla policy $\pi$.

**Best-of-N (BoN) Sampling.** As a common and efficient strategy to sample multiple reasoning trajectories from LLMs, Best-of-$N$ sampling selects the trajectory with maximal reward among $n$ independent rollouts from $\pi_0$ to enhance policy performance. Formally, given candidate actions $\{a^{(i)}\}_{i=1}^n \sim \pi_0(\cdot|s)$, the chosen action is $a^* = \arg\max_{a^{(i)}} Q(s, a^{(i)})$. This strategy effectively leverages the exploration-exploitation trade-off through parallel sampling (Gao et al., 2023; Go et al., 2023).

To quantify the distributional divergence induced by BoN sampling, prior work (Hilton & Gao, 2022; Scheurer et al., 2023; Coste et al., 2023) has analyzed the KL divergence between the BoN distribution $\pi_{\mathrm{BoN}}$ and the original policy $\pi$. For continuous trajectory spaces $\mathcal{S}$, the corresponding KL divergence is given by $\mathrm{KL}(\pi_{\mathrm{BoN}} \| \pi) = \log n - \frac{n-1}{n}$.

### 2.2 Learning from Positive Samples

Building upon the reward equivalence principle stated in Eq. A1, we first formalize a key probabilistic characteristic of BoN sampling:

**Lemma 2.1.** *Let $\pi(\theta, s)$ be a distribution over parameters $\theta$ and trajectory $s$, where each $s$ is associated with a binary reward $R(s) \in \{0, 1\}$. Define $p \triangleq \mathbb{E}_{s \sim \pi(\theta, \cdot)}[R(s) = 1] > 0$. Consider the*

*BoN sampling: $n = n_0 \to \infty$ and sample $\{s_1, s_2, \ldots, s_n\}$ i.i.d. from $\pi_\theta$. BoN selects $s^*$ uniformly from the subset with $R(s_i) = 1$. We have that, The probability of selecting $s^*$ is converge to $\frac{\pi(\theta, s)}{p}$, which is independent of n.*

The proof follows directly from the union law of BoN sampling ($\text{BoN}_{n+m} = \text{BoN}_2(\text{BoN}_m, \text{BoN}_n)$) and the trivial distinguishability of $0 - 1$ rewards. This result reveals that for problems with attainable positive responses, we are using a BoN generator with an arbitrary sampling budget to construct the positive training samples.

BoNBoN (Gui et al., 2024) empirically shows that BoN sampling achieves the optimal win rate under fixed KL constraint by exhaustive search over the positive support. Therefore, the behavior cloning on BoN-selected positive samples directly learns the analytic solution to Eq. 1. Intuitively, since every correct answer is preferred identically in the outcome-supervised sense, we only need to sample until we get a positive example, whose generating probability distribution will be the same as randomly picking from arbitrarily large numbers of samples.

Based on the theoretical understanding established, we formulate the first component of the learning objective in OREAL by incorporating KL-constrained max-likelihood-objective over positive examples obtained through sampling:

$$\mathcal{L}_1(\theta) = \underbrace{\mathbb{E}_{s \sim \mathcal{D}^+}\left[-\log \pi_\theta(s)\right]}_{\text{Positive example alignment}} + \beta \underbrace{\text{KL}(\pi_\theta \parallel \pi_{\text{old}})}_{\text{Policy constraint}},$$

where $\mathcal{D}^+$ denotes the set of positive trajectories selected via BoN sampling from RL rollouts.

## 2.3 Learning from Negative Samples

As established in Section 2.2, direct behavioral cloning on positive responses can effectively recover the policy distribution. BOND (Sessa et al., 2024) proposes estimating Jeffreys divergence (Jeffreys, 1946) for the BoN strategy to train with both positive and negative samples, and demonstrates that signals from unsuccessful trajectories provide critical information about decision boundaries and failure modes.

In this section, we will discuss the relationship between the BoN (Best-of-N) distribution and the optimization objective defined in Eq. 1, then elucidate the necessity of reward reshaping when training with negative samples. Specifically, the transformed BoN distribution can be expressed as

$$\pi_{\text{bon}}(s) = \pi(s)\left[R(s) \cdot \frac{1 - (1-p)^n}{p} + (1 - R(s)) \cdot (1-p)^{n-1}\right]. \tag{2}$$

Consider a scenario where two correct and two incorrect solutions are sampled, yielding an empirical accuracy of 50%. However, the probability of selecting negative samples under Best-of-4 becomes $(0.5)^4 = 6.25\%$, significantly lower than the original distribution. This discrepancy necessitates reward shaping to maintain consistency between our optimization target and the expected return under the BoN distribution.

Building on BoN-RLB's (Chow et al., 2024) application of the log-likelihood trick for BoN-aware policy gradients, we analyze the reward shaping technic for negative samples to maintain gradient consistency with Section 2.2. With expectation return $p$ follow the definition in Lemma 2.1. We derive the gradient components as:

$$\mathbb{E}_{s \sim \pi_{\text{bon}}}\left[\nabla_\theta\left(I_{D_+}(s)\log\pi(s)\rho(D_+)\right)\right] = \frac{1}{1-p}\mathbb{E}_{s \sim \pi_{\text{bon}}}\left[\nabla_\theta\left(I_{D_-}(s)\log\pi(s)\rho(D_-)\right)\right],$$

where $I_{D_+}(s)$ and $I_{D_-}(s)$ denote indicator functions for positive and negative sample sets respectively, and $\rho(D) \triangleq \frac{\mathbb{E}_{s \sim \pi_{\text{bon}}}[I_D(s)]}{p_D}$ are their density. Notably, these indicators are independent of policy parameters $\theta$.

This derivation reveals that gradient consistency requires reshaping negative sample rewards to $R^\star(s) \triangleq (1 - p)R(s)$. Based on this reward shaping, we can construct policy optimization both on positive and negative samples for optimal policy. In this paper, we apply a similar setting to RLOO (Fukunaga & Hummels, 1989), namely $R_{RLOO}(s) = R(s) - \frac{1}{N-1}\sum_{s^\star \neq s} R(s^\star)$ for unbiased mean reward and train with policy gradient. The second part of our OREAL objective is then formulated as below:

$$\mathcal{L}_2(\theta) = \mathbb{E}_{s \sim S_-}\left[F(1-p) \cdot \log \frac{\pi_\theta(s)}{\pi_{old}(s)}\right] + \beta \text{KL}(\pi_\theta \parallel \pi_{\text{old}}),$$

where $p = \mathbb{P}_{\theta \sim \pi}[R(\theta) = 1]$, $S_-$ is the failed subset generated by policy model, and $F$ represents the preprocessing for advantage scores to serve as a generalized form, for example, $F(1-p) \triangleq \frac{r_i - mean(\{r_i...r_n\})}{std(\{r_i...r_n\})}$ in the recent GRPO (Shao et al., 2024) algorithm, where $mean(\{r_i...r_n\}) \to p$ when $n \to \infty$.

## 2.4 Dealing with Long Reasoning Chains

In the previous discussion, we introduced the adaptation of binary reward training in response space. However, since the outcome supervision only provides feedback at the sequence level, this modeling essentially reduces to a contextual bandit without internal reward modeling within the MDP. We therefore choose to use some low-cost alternatives for sequence-level reweighting.

Since intermediate rewards are not provided in mathematical reasoning tasks, we define an advantage function based solely on outcome feedback:

$$A(s_{\leq t}) = V^\pi(s_{\leq t+1}) - V^\pi(s_{\leq t}). \tag{3}$$

This formulation treats $A(s_{\leq t})$ as a token-wise credit assignment mechanism, estimating each token's contribution toward the final outcome.

For a pair of responses $y_1$ and $y_2$ to the same query, their initial values coincide $V_0^1 = V_0^2$. The win rate $p(y_1 > y_2)$ between them then satisfies:

$$\sigma(r(y_1) - r(y_2)) = \sigma\left(\left(V_0^1 + \sum_{t=0}^T \gamma^t A_{y_1}^t\right) - \left(V_0^2 + \sum_{t=0}^T \gamma^t A_{y_2}^t\right)\right)$$
$$= \sigma\left(\sum_{t=0}^T \gamma^t \left(A_{y_1}^t - A_{y_2}^t\right)\right). \tag{4}$$

Equation 4 indicates that for any function family $\mathcal{A} = \{A(s_{\leq t})\}$, a cumulative reward function through sequence aggregation can be constructed to model rewards: $r^*(s) \triangleq \sum_{t=0}^T \gamma^t A(s_{\leq t})$, which is trainable via preference pairs $\{(y_w, y_l)\}$ by fitting the outcome feedback.

The learned $A(s_{\leq t})$ then serves as a weighting function for credit assignment, which is used to reweight the original training loss, emphasizing critical reasoning steps or errors. Following the practice from (Cobbe et al., 2021), we directly train a token-level reward function $w(s_{\leq t})$ satisfying

$$\frac{1}{T}\sum_{t=0}^T w(s_{\leq t}) = r(s),$$

without constraining KL-divergence to reference model in reward model training. These sequential rewards can serve as a proxy for the contribution of thinking steps to the result accuracy. We further discuss the training details of this model and analyze the visualization of its token-wise scoring effects later in Appendix D.

In practice, we decompose the output weight $w(s)$ for positive and negative samples and clip on the positive axis to prevent reversing the direction of the optimized gradient, denoted as $\omega^+ = \max(2\sigma(w) - 1, 0)$ and $\omega^- \max(1 - 2\sigma(w))$.

With input query $d$, the overall loss is as follows:

$$
\begin{aligned}
\mathcal{L}_{\text{total}}(d) \triangleq \mathbb{E}_{s \sim S} &\left[ \sum_{t=0}^{T} \left( -\omega_{s \leq t}^{+} \log \pi_{\theta}(s_{\leq t}|d) I_{D_+}(s) + \eta \, \omega_{s \leq t}^{-} \log \frac{\pi_{\theta}(s_{\leq t}|d)}{\pi_{old}(s_{\leq t}|d)} I_{D_-}(s) \right) \right] \\
&+ \beta \mathrm{KL}(\pi_{\theta}(\cdot|d) \parallel \pi_{\text{old}}(\cdot|d)),
\end{aligned}
\tag{5}
$$

where $\eta$ represents the balancing weights for positive and negative losses.

## 3 Implementation

### 3.1 Policy Initialization

We utilize Qwen2.5-7B and Qwen2.5-32B (Yang et al., 2024) as the base model. Initially, we fine-tune the base models using long chain-of-thought data obtained through rejection sampling (Yuan et al., 2023). This rejection sampling fine-tuned (RFT) (Yuan et al., 2023) models then serve as the initialization for the policy model in our RL framework. We also explore to use of DeepSeek-R1-Distill-Qwen-7B (DeepSeek-AI et al., 2025) as the initial policy model and perform OREAL on it and discuss the influence of different initial policy models in Appendix C. The training data for the RFT models consists of in-house datasets supported by OpenDataLab (He et al., 2024b) and open-source datasets including Numina (Li et al., 2024) and the training set of MATH (Hendrycks et al., 2021). In addition, we implement a skill-based enhancement approach to further enhance the capability of the model, whose details can be found in Appendix C.1. We report the performance of the initial policy models and discuss the impact of policy initialization in Appendix C.

### 3.2 Reinforcement Learning

**Data Preparation.** During the on-policy RL process, we utilize questions from Numina, MATH training sets, and historical AMC/AIME (without AIME2024) competitions. For each question, we independently sample 16 trajectories from the RFT models. To increase the difficulty of training queries, only questions with average correctness rates between 0 and 0.8 are retained for further training.

**Outcome Reward Signal.** We employ the Qwen2.5-72B-Instruct (Yang et al., 2024) as a generative verifier, combined with a rule-based verifier, to evaluate the correctness of the outputs and provide binary rewards. This combination enhances the robustness of correctness assessment, mitigating issues like the false negative of the rule-based verifier.

**Token-level Reward Model.** We directly use the binary outcome rewards and optimize using the cross-entropy loss. Details can be found in Appendix D.

**Training Algorithm.** The loss function for the policy model follows the formulation described in Section 2. The complete RL training procedure is described in Algorithm 1.

**Model Training Settings.** The policy model is initialized from the RFT model. Similarly, the token-level reward model is also initialized with the same weights, but its output layer is replaced with a linear layer that produces a one-dimensional scalar. The weights of this layer are initialized to zero to ensure unbiased fine-grained token re-weighting at the start of training. The training hyperparameters can be found in can be found in Appendix B.

## 4 Experiment

### 4.1 Evaluation Setup

**Baseline.** We conduct evaluations against several baselines, including GPT-4o-0513 (OpenAI, 2024a), Claude-Sonnet-3.5-1022 (Anthropic, 2024), OpenAI-o1-mini, OpenAI-o1-preview (OpenAI, 2024b), Qwen2.5-Instrust-7B, Qwen2.5-Math-Instrust-7B, Qwen2.5-

---

**Algorithm 1** The OREAL Reinforcement Learning Algorithm

---

1: **Inputs:** Question set $\mathcal{D}$, policy model $\pi_\theta$, token-level reward model $w_\theta$, number of iterations $N$, batch size $B$, number of rollouts per question $K$.
2: Initialize policy $\pi_0$ and token-level reward model $w_0$ with $\pi_{\text{sft}}$.
3: **for** $i = 0, \dots, N$ **do**
4:  Sample a batch of questions $\mathcal{D}_i \subseteq \mathcal{D}$ of size $B$.
5:  For $x \in \mathcal{D}_i$, generate $K$ policy samples: $Y = \{y_1, \dots, y_K\}$ where $y_k \sim \pi_i(x)$
6:  Obtain binary rewards $\{r_1, \dots, r_K\}$ from verifier.
7:  Compute correctness rate: $p = \frac{1}{K} \sum_{k=1}^{K} r_k$ for reward shaping.
8:  Retain questions with $0 < p < 1$ to avoid trivial cases.
9:  Select one correct $y^+$ and one incorrect sample $y^-$ for each question to avoid imbalance between positive and negative samples.
10:  Performing fine-grained re-weighting of each token with $w_i$.
11:  Use Eq (A2) to update $w_i$.
12:  Update $\pi_i$ with Eq (5)
13: **end for**
14: **Return:** The optimized policy model $\pi^*$.

---

Instrust-32B (Yang et al., 2024), QwQ-32B-Preview (Team, 2024), DeepSeek-R1-Distill-Qwen-7B, DeepSeek-R1-Distill-Qwen-32B (DeepSeek-AI et al., 2025), SimpleRL (Zeng et al., 2025), PRIME (Cui et al., 2025), rStarMath (Guan et al., 2025). For part of the baseline, we directly use the results from their report, which we mark with *.

**Benchmark.** We use some well-established mathematical datasets for evaluation, including MATH-500 (Hendrycks et al., 2021), AIME2024 (MAA), AIME2025 (Part1) (MAA), LiveMathBench (Liu et al., 2024c), and OlympiadBench (He et al., 2024a).

**Metrics.** We use pass@1 as the metric for evaluation under the zero-shot chain-of-thought setting and use greedy decoding for each sample to assess correctness.

### 4.2 Overall Results

Tabel 1 shows the results of the comprehensive evaluation, highlighting the performance of our proposed models across different parameter scales. Notably, at the 7B scale, OREAL-7B achieves a remarkable pass@1 accuracy of 91.0 on the MATH-500 and 59.9 on Olympiad-Bench. To the best of our knowledge, this is the first time a model of this size has reached such a high level of accuracy using RL instead of distillation. This performance not only establishes a new milestone for RL-based methods but also surpasses significantly larger models, including QwQ-32B-Preview and OpenAI-o1-mini, demonstrating the effectiveness of our approach. Furthermore, after applying OREAL on the previous best 7B model, DeepSeek-R1-Distill-Qwen-7B, the resulting model, OREAL-DSR1-Distill-Qwen-7B, obtains 94.0 and 66.1 pass@1 accuracy on MATH-500 and OlympiadBench, respectively, setting new records among all 7B models. This result verifies the effectiveness of OREAL even when faced with strong initial policies.

For 32B models, OREAL-32B achieves a groundbreaking pass@1 accuracy of 95.0 on MATH-500, 46.7 on AIME2025-I, 74.8 on LiveMathBench, and 72.4 on OlympiadBench, setting a new state-of-the-art among all previously reported models. These results underscore the advantages of our methodology, including its scalability for training superior mathematical reasoning models across different model sizes.

Compared to the most competitive baseline, DeepSeek-R1-Distill-Qwen series, OREAL-32B demonstrates a clear advantage, whereas OREAL-7B lags slightly behind than the distilled 7B model, despite being trained on the same dataset as OREAL-32B. We attribute this discrepancy to the different affinities of the base models for the post-training data. Qwen-7B and Qwen-32B may exhibit varying degrees of knowledge gaps due to model sizes and pre-training settings. Our training data appears to better complement the existing knowledge of Qwen-32B, while it may be less effective in bridging gaps for Qwen-7B.

| Model | MATH-500 | AIME2024 | AIME2025-I | LiveMath | Olympiad |
|---|---|---|---|---|---|
| API Models | | | | | |
| GPT-4o-1120 | 72.8 | 16.7 | 13.3 | 44.8 | 33.7 |
| Claude-3.5-Sonnet-1022 | 78.3 | 13.3 | 3.3 | 46.7 | 35.4 |
| OpenAI-o1-preview | 85.5 | 44.6 | 40.0 | 71.0 | 43.6 |
| OpenAI-o1-mini | 90.0 | 56.6 | 46.7 | 74.4 | 46.3 |
| 7B Models | | | | | |
| Qwen2.5-Instrust-7B | 76.6 | 13.3 | 0.0 | 37.0 | 29.1 |
| Qwen2.5-Math-Instrust-7B | 81.8 | 20.0 | 13.3 | 44.1 | 31.1 |
| rStar-Math-7B | 78.4* | 26.7* | - | - | 47.1* |
| Qwen2.5-7B-SimpleRL | 82.4* | 26.7* | - | - | 37.6* |
| Eurus-2-7B-PRIME | 79.2* | 26.7* | - | - | 42.1* |
| DeepSeek-R1-Distill-Qwen-7B | 92.8* | **55.5*** | **40.0** | **65.6** | 64.1 |
| **OREAL-7B** | 91.0 | 33.3 | 33.3 | 62.6 | 59.9 |
| **OREAL-DSR1-Distill-Qwen-7B** | **94.0** | 50.0 | **40.0** | **65.6** | **66.1** |
| 32B Models | | | | | |
| Qwen2.5-Instrust-32B | 80.6 | 20.0 | 13.3 | 50.8 | 40.4 |
| QwQ-32B-Preview | 90.6 | 50.0 | 40.0 | 72.7 | 58.5 |
| DeepSeek-R1-Distill-Qwen-32B | 94.3* | **72.6*** | **46.7** | 67.7 | 71.2 |
| **OREAL-32B** | **95.0** | 60.0 | **46.7** | **74.8** | **72.4** |

Table 1: Overall evaluation results for OREAL and each baseline. "OREAL-DSR1-Distill-Qwen-7B" denotes the DeepSeek-R1-Distill-Qwen7B trained by OREAL. "AIME2025-I", "LiveMath" and "Olympiad" represent "AIME 2025 Part1", "LiveMathBench", and "OlympiadBench", respectively. For models at the parameter scale of 7B and 32B, we use Bold and Underlined to represent the best and second best performance, respectively. For part of the baseline, we directly use the results from their report, marked with *.

| Setting | MATH-500 |
|---|---|
| Initial Policy | 84.8 |
| + REINFORCE (baseline) | 85.8 |
| + Reward Shaping | 86.6 |
| + Behavior Cloning | 87.6 |
| + Token Re-weighting | 89.0 |
| + Skill-based Enhancement | 91.0 |

Table 2: Ablation study for 7B models performance on MATH-500 with different reinforcement learning settings.

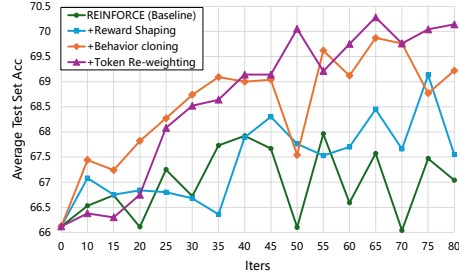

Figure 2: Average test accuracy of 7B models across different training steps.

In addition, OREAL-DSR1-Distill-Qwen-7B improves the MATH-500 score from 92.8 to 94.0 and also achieves gains on LiveMathBench and OlympiadBench. However, its performance on the AIME benchmark series is comparatively weaker. We observe the same disadvantages of OREAL-32B and OREAL-7B, whose AIME2024 scores are relatively lower than the best scores. Since the overall performance verifies the effectiveness of the OREAL algorithm, we attribute the reason to the deficiency (*e.g.*, in response quality, query difficulty, and quantity) of RFT data and RL training queries for obtaining high performance in the domain of AIME and leave it for the future work.

### 4.3 Ablation Study

To verify the effectiveness of each component described in Section 2, we progressively add the proposed component based on the 7B model and compare the evaluation results on MATH-500, starting from REINFORCE (Sutton et al., 1999) as baseline.

As shown in Tabel 2, we add each component step by step, where "Reward Shaping" represents $L_2$ introduced in Section 2.3, "Behavior Cloning" represents $L_1$ introduced in Section 2.2, "Token Re-weighting" represents $L_{total}$ introduced in Section 2.4. The gradual addition of the modules steadily increases the Pass@1 scores of the 7B model on MATH-500, proving the effectiveness of our method. Ultimately, the policy model is raised from an initial score of 84.8 to 91.0.

We also report average pass@1 accuracy across all benchmarks during the training process with different RL settings. As shown in Figure 2, the REINFORCE training process is unstable, which can be mitigated by "Reward Shaping". "Behavioral Cloning" for positive samples can speed up convergence and show better performance early in training. Although the performance growth of "Token Re-weighting" is relatively slow in the early stage of training, it ultimately obtains the best results.

## 5 Related Work

**Stimulate Reasoning using Chain of Thought.** In mathematical reasoning tasks, Chain of Thought (CoT) (Wei et al., 2022) is recognized as a crucial technique to enhance the reasoning ability of large language models (LLMs), which can be implemented through few-shot examples (Wei et al., 2022) or prompt engineering (Kojima et al., 2022). Self-consistency (Wang et al., 2022) is further proposed to generate and voting through multiple CoTs. In addition to simple CoTs, various search methods have been explored that simultaneously consider multiple potential CoTs, such as Tree-of-Thought (ToT) (Yao et al., 2024) and Graph-of-Thought (GoT) (Besta et al., 2024), which extend the idea to tree or graph structure, offering more flexibility in developing CoTs and backtracking. However, these methods mainly stimulate the reasoning capability of LLMs by prompts without parameter updates, these inference-time techniques do not fundamentally improve the underlying ability of LLMs.

**Reasoning Enhancement by Supervised Fine-tuning.** To let the LLMs essentially acquire the reasoning abilities, many studies Ying et al. (2024); Yu et al. (2023); Liu et al. (2024b); Li et al. (2023); Liu & Low (2023); Yue et al. (2023) have explored synthesizing high-quality data to conduct supervised fine-tuning (SFT) on LLMs. But this method heavily relies on high-quality training data and a existing high-performing model (Lightman et al., 2023). As a result, many existing works (Huang et al., 2024; Min et al., 2024) have turned to distilling knowledge from powerful, large-scale models to synthesize data, yielding good results. However, distilled-based methods receive the limitations of the teacher model. Some studies argue that such approaches merely transform the model into a knowledge retriever, rather than an actual reasoner (Kambhampati, 2024).

**Reinforcement Learning for LLM.** Compared to SFT, reinforcement learning (RL) offers better generalization and is therefore considered a more fundamental training aproach (Chu et al., 2025). The advent of the o1 family of models (OpenAI, 2024b) and a series of o1-like works (DeepSeek-AI et al., 2025; Zeng et al., 2025; Cui et al., 2025; Guan et al., 2025) make the importance of large-scale RL for inference became more apparent. Currently, the mainstream approach to RL involves using outcome reward signals (DeepSeek-AI et al., 2025; Zeng et al., 2025; Kazemnejad et al., 2024) and there are different views in the community on how to use that reward signal. ReST$^{EM}$ (Singh et al., 2023) and RFT (Yuan et al., 2023) simply select the positive samples based on the binary signal and only use them for behavior cloning. GRPO (Shao et al., 2024), RLOO (Fukunaga & Hummels, 1989; Ahmadian et al., 2024), REINFORCE (Sutton et al., 1999), use both positive and negative samples for policy updating, but facing the challenges of sparse reward in long sequence. PPO (Schulman et al., 2017) makes the preference modeling on sequence-level. Different from them, to explore the limit of outcome reward, OREAL presents a unified framework, grounded in the unique characteristics of mathematical reasoning tasks.

## 6 Conclusion

This paper aims to explore the limit of **O**utcome **REw**A**r**d-based reinforcement **L**earning for mathematical reasoning tasks, and proposes a unified policy optimization framework,

termed OREAL, grounded in three key insights: 1) Behavior cloning on positive trajectories from Best-of-n (BoN) sampling is both necessary and sufficient for optimal policy learning under binary feedback; 2) Accordingly, a reward-shaping mechanism should be further introduced to transform the reward function derived from the optimal policy; 3) An efficient token-level credit assignment scheme can be achieved through trajectory advantage decomposition without relying on additional value networks. Together, these components form a theoretically grounded, general, and scalable approach for mathematical reasoning tasks. With OREAL, we are the first to improve the performance of a 7B model on the MATH-500 accuracy to 91 using the RL method instead of distillation, which even surpasses OpenAI-o1-mini and QwQ-32B-Preview. Even when taking the previous best 7B model, DeepSeek-R1-Distill-Qwen-7B, as initial policy, OREAL can improve it to 94 accuracy on MATH-500, being even on par with the previous best 32B models. OREAL-32B also obtains new state-of-the-art results among the 32B model on MATH-500, LiveMathBench, and OlympiadBench.

Along with the experimental observations presented in this paper, we also find two factors that are crucial for the success of scalable RL for mathematical reasoning tasks, which become the primary focus of our future work. First, the initial policy model should be as free of knowledge deficiencies as possible, as this serves as the foundation for further improvement during the RL stage. A strong starting point ensures that RL can effectively and efficiently incentivize the underlying capability of LLMs obtained through pre-training or supervised fine-tuning. Towards this goal, it is a practical way to conduct distillation or data synthesis with DeepSeek-R1 or DeepSeek-V3, which is not explored in this work as it is orthogonal to our investigation. Second, the quality of the data used in the RL phase must be diverse and sufficient in terms of difficulty, quantity, and scope. A well-balanced dataset enables the model to reach its full potential by exposing it to a broad range of challenges and learning opportunities. Thus, we believe it is still valuable to make efforts in the pre-training and post-training data construction process.

## Acknowledgement

We thank the anonymous reviewers and area chair for their helpful comments. This project is funded in part by the Hong Kong Generative AI Research and Development Center (HKGAI) under the Innovation and Technology Commission (ITC)'s InnoHK. Dahua Lin is a PI of HKGAI under the InnoHK. This project is also supported by the Shanghai Artificial Intelligence Laboratory.

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

## A Binary Feedback under Outcome Supervision.

Though a reasoning trajectory usually contains multiple reasoning steps with thousands of tokens, there lacks an efficient approach to automatically label the correctness of each token or reasoning step in math reasoning tasks. Thus, a practical way is to parse the final answer from the reasoning trajectory (DeepSeek-AI et al., 2025; Lambert et al., 2024), evaluate its correctness based on rules or models, and then provide an outcome reward at the end of the trajectory as below:

$$R(s_t) = \begin{cases} 1 & \text{if } t \text{ is the end step and the answer is correct} \\ 0 & \text{otherwise,} \end{cases} \tag{A1}$$

which intrinsically treats the correct trajectory equally for learning. Moreover, the reward signal is severely sparse when compared to the thousands of tokens and does not provide any signal of progress or correctness for intermediate steps. The resulting reward distribution of trajectories is also different from that of the dense reward function constructed through preference pairs in traditional RL for large language models (Ouyang et al., 2022), which induces a more appropriate optimization framework for mathematical reasoning tasks, discussed in the next section.

## B Hyperparameters Details

During training iterations, each batch consists of 64 questions, with 16 rollouts per question. The max length of each rollout trajectory is set to 16384 tokens. Then the correctness of each response is averaged to calculate the pass rate, and questions with an overall pass rate of 0 or 1 are discarded. For the remaining trajectories, we retain only one correct response and one incorrect response per question, ensuring a balanced distribution of positive and negative samples for token-level reward model training.

For optimization, the policy model is trained with a learning rate of $5e{-}7$, while the token-level reward model is trained with a learning rate of $2e{-}6$. The latter undergoes a 10-step warm-up phase before training begins. Both models employ a cosine annealing learning rate schedule, decaying to $1/5$ of the initial learning rate over time. We optimize both models using the AdamW optimizer. The total number of training steps is 80, with evaluation conducted every 10 steps. The KL coefficient $\beta$ is set to 0.01. We select the best-performing model determined by evaluation metrics.

## C Analysis of the Impact of Initial Policy Models

### C.1 Skill-based Enhancement

During the RL training procedure, we observe that the model consistently struggles with certain types of questions, particularly those involving specific knowledge and skill areas, such as trigonometric constant transformations, probability statistics, series transformations, *etc*. We believe this is caused by the insufficient learning of the base model on these concepts in the Pre-training or RFT stages.

To address this problem, we implement a skill-based enhancement approach, using the MATH dataset to reduce the high cost of skill annotation. Specifically, we annotate each question in the training set with its corresponding core skill. For questions that the model repeatedly fails to answer correctly during the RL phase, we perform data augmentation by including similar questions from the training set that share the same skill. These augmented questions are then added to the training data during the RFT stage to help the model better internalize these skills.

### C.2 Results

We further analyze OREAL by adopting it to several different initial policy models, as shown in Table A1. OREAL consistently improves the performance of each initial policy

| Model | MATH-500 | AIME2024 | AIME2025-I | LiveMath | Olympiad |
|---|---|---|---|---|---|
| OREAL-7B-SFT-wo-enhance | 84.8 | 26.7 | 26.7 | 55.0 | 55.1 |
| OREAL-7B-wo-enhance | 89.0 | 36.7 | 40.0 | 60.1 | 58.1 |
| OREAL-7B-SFT | 86.4 | 26.7 | 26.7 | 54.2 | 56.0 |
| OREAL-7B | 91.0 | 33.3 | 33.3 | 62.6 | 59.9 |
| DeepSeek-R1-Distill-Qwen-7B | 92.8* | 55.5* | 40.0 | 65.6 | 64.1 |
| OREAL-DSR1-Distill-Qwen-7B | 94.0 | 50.0 | 40.0 | 65.6 | 66.1 |
| OREAL-32B-SFT | 92.6 | 43.3 | 46.7 | 71.9 | 68.7 |
| OREAL-32B | 95.0 | 60.0 | 46.7 | 74.8 | 72.4 |

Table A1: Evaluation for the performance of OREAL on different initial policy models. Here, "-SFT" and "DeepSeek-R1-Distill-Qwen7B" denote the initial policy model. "wo-enhance" means the model which do not perform the skill-based enhancement during the SFT stage.

model, including our own trained model and the strong distilled model (DeepSeek-AI et al., 2025), on MATH-500, LiveMathBench, and OlympiadBench, except slight fluctuations on AIME2024 and AIME2025 part1 when the performance of initial policy models are already high (*e.g.*, DeepSeek-R1-Distill-Qwen-7B), which demonstrates the generality of OREAL.

After adding skill-based enhancement data, there is a significant rise in MATH-500 scores for the initial policy model (row 1 and row 3) and the corresponding RL-trained model (row 2 and row 4). Since our enhancement is performed primarily for the MATH-500, this verifies the effectiveness of the skill-based enhancement approach. In addition, the performance of the model after RL is strongly correlated with the capabilities of the initial policy model itself. The stronger the initial policy model, the higher the performance that RL can deliver, indicating the importance of policy initialization.

# D    Token Level Reward Model

## D.1    Training Settings

For the token-level reward model, we directly use the binary outcome rewards provided by the verifier and optimize using the cross-entropy loss:

$$\mathcal{L}_{\text{CE}} = -\mathbb{E}_{(s,r)\sim\mathcal{D}}\left[r\log p(s) + (1-r)\log(1-p(s))\right], \tag{A2}$$

where $s$ represents the sampled trajectory, $r \in \{0,1\}$ is the binary outcome reward from the verifier, and $p(s) = \sigma(\frac{1}{T}\sum_{t}^{T} w(s_t))$ denotes the predicted probability of correctness by the token-level reward model $w$.

To further analyze the behavior of the token-level reward model, we visualize its output distribution $w(s_t)$ during the on-policy RL training process (see Appendix D.2). In this training paradigm, $w(s_t)$ assigns token-wise importance scores across the chain-of-thought reasoning process, capturing each token's contribution to the final correctness of the generated response. Consequently, this allows us to leverage $w(s_t)$ for importance sampling during the optimization process, enabling a more principled selection of informative tokens.

## D.2    Score Visualization

Figure A1 and A2 show the token-level reward model scores across responses. The values are normalized to [0, 1]. Cooler colors indicate higher reward scores, while warmer colors denote lower scores. For correct responses, the overall REWARDS are high, especially at the end, although there are a few lower sections in the middle. For incorrect responses, the distribution of rewards is reversed, and the closer to the end the lower the rewards. This indicates that not all tokens contribute to the response equally and it is important to assign token-level credits to the sequences.

To solve the equation $( z^4 + 4z^3i - 6z^2 - 4zi - i = 0 )$, we start by recognizing that it can be rewritten in a form that suggests a binomial expansion. Notice that the coefficients of the polynomial match the binomial coefficients of $( (z + i)^4 )$ except for the constant term. Specifically, $[ (z + i)^4 = z^4 + 4z^3i + 6z^2i^2 + 4zi^3 + i^4 = z^4 + 4z^3i - 6z^2 - 4zi + 1. ]$Thus, we can rewrite the given equation as $[ (z + i)^4 - 1 - i = 0 \quad \text{or} \quad (z + i)^4 = 1 + i. ]$Next, we need to express $( 1 + i )$ in polar form. The magnitude of $( 1 + i )$ is $[ |1 + i| = \sqrt{1^2 + 1^2} = \sqrt{2}, ]$and the argument is $[ \arg(1 + i) = \frac{\pi}{4}. ]$Therefore, we can write $[ 1 + i = \sqrt{2} \left( \cos \frac{\pi}{4} + i \sin \frac{\pi}{4} \right). ]$Now, we need to find the fourth roots of $( \sqrt{2} \left( \cos \frac{\pi}{4} + i \sin \frac{\pi}{4} \right) )$. The fourth roots are given by $[ (z + i) = \left( \sqrt{2} \right)^{1/4} \left( \cos \frac{\frac{\pi}{4} + 2k\pi}{4} + i \sin \frac{\frac{\pi}{4} + 2k\pi}{4} \right) = 2^{1/8} \left( \cos \frac{\pi + 8k\pi}{16} + i \sin \frac{\pi + 8k\pi}{16} \right), ]$for $( k = 0, 1, 2, 3 )$. These roots are $[ z + i = 2^{1/8} \left( \cos \frac{\pi}{16} + i \sin \frac{\pi}{16} \right), ]$ $[ z + i = 2^{1/8} \left( \cos \frac{9\pi}{16} + i \sin \frac{9\pi}{16} \right), ]$ $[ z + i = 2^{1/8} \left( \cos \frac{17\pi}{16} + i \sin \frac{17\pi}{16} \right), ]$ $[ z + i = 2^{1/8} \left( \cos \frac{25\pi}{16} + i \sin \frac{25\pi}{16} \right). ]$Thus, the solutions $( z )$ are $[ z = 2^{1/8} \left( \cos \frac{\pi}{16} + i \sin \frac{\pi}{16} \right) - i, ]$ $[ z = 2^{1/8} \left( \cos \frac{9\pi}{16} + i \sin \frac{9\pi}{16} \right) - i, ]$ $[ z = 2^{1/8} \left( \cos \frac{17\pi}{16} + i \sin \frac{17\pi}{16} \right) - i, ]$ $[ z = 2^{1/8} \left( \cos \frac{25\pi}{16} + i \sin \frac{25\pi}{16} \right) - i. ]$These solutions represent the vertices of a square in the complex plane, each with a distance of $( 2^{1/8} )$ from the point $( -i )$. The side length of the square is $( 2^{1/8} \sqrt{2} = 2^{5/8} )$. The area of the square is $[ \left( 2^{5/8} \right)^2 = 2^{5/4}. ]$Thus, the area of the polygon is $( 2^{5/4} )$, and we have $( p = 2 )$, $( a = 5 )$, and $( b = 4 )$. Therefore, $( a + b + p = 5 + 4 + 2 = 11 )$.The final answer is $[ \boxed{11}. ]$

Figure A1: Token-level reward model score visualization for a correct response.

Given the equation $ z^4 + 4z^3i - 6z^2 - 4zi - i = 0 $, we aim to find the area of the convex polygon formed by the solutions in the complex plane.First, we rewrite the equation in a more manageable form: $$ z^4 + 4iz^3 - 6z^2 - 4iz - i = 0 $$We will use the substitution $ w = z + i $. Then, $ z = w - i $, and substituting $ z $ into the equation, we get: $$ (w - i)^4 + 4i(w - i)^3 - 6(w - i)^2 - 4i(w - i) - i = 0 $$Expanding each term: $$ (w - i)^4 = w^4 - 4w^3i + 6w^2i^2 - 4wi^3 + i^4 = w^4 - 4w^3i - 6w^2 - 4wi + 1 $$ $$ 4i(w - i)^3 = 4i(w^3 - 3w^2i + 3wi^2 - i^3) = 4i(w^3 - 3w^2i - 3w + i) $$ $$ -6(w - i)^2 = -6(w^2 - 2wi + i^2) = -6(w^2 - 2wi - 1) $$ $$ -4i(w - i) = -4i(w - i) $$ $$ -i = -i $$Combining all these terms: $$ w^4 - 4w^3i - 6w^2 - 4wi + 1 + 4i(w^3 - 3w^2i - 3w + i) - 6(w^2 - 2wi - 1) - 4i(w - i) - i = 0 $$Simplifying: $$ w^4 - 4w^3i - 6w^2 - 4wi + 1 + 4iw^3 - 12iw^2i - 12iw + 4i^2 - 6w^2 + 12wi + 6 - 4iw + 4i^2 - i = 0 $$ $$ w^4 - 4w^3i - 6w^2 - 4wi + 1 + 4iw^3 - 12iw^2 - 12iw + 4 - 6w^2 + 12wi + 6 - 4iw + 4 - i = 0 $$ $$ w^4 - 4w^3i + 4iw^3 - 6w^2 - 6w^2 - 12iw^2 - 4wi + 12wi - 4iw - i + 1 + 4 + 6 + 4 = 0 $$ $$ w^4 - 4w^3i + 4iw^3 - 12w^2 - 12iw^2 - i + 15 = 0 $$This equation is complex and solving it directly is challenging. Instead, we recognize that the solutions form a symmetric polygon. By symmetry and properties of polynomial roots, the solutions are likely to be symmetric about the real axis.Given the symmetry and the nature of the polynomial, we can infer that the solutions form a square. The vertices of the square are at $ \pm 1 \pm i $.The side length of the square is: $$ \sqrt{(1 - (-1))^2 + (i - (-i))^2} = \sqrt{2^2 + 2^2} = \sqrt{8} = 2\sqrt{2} $$The area of the square is: $$ (2\sqrt{2})^2 = 8 $$Expressing 8 in the form $ p^{a/b} $: $$ 8 = 2^3 = 2^{3/1} $$Thus, $ a = 3 $, $ b = 1 $, and $ p = 2 $. Therefore, $ a + b + p = 3 + 1 + 2 = 6 $. $$ \boxed{6} $$

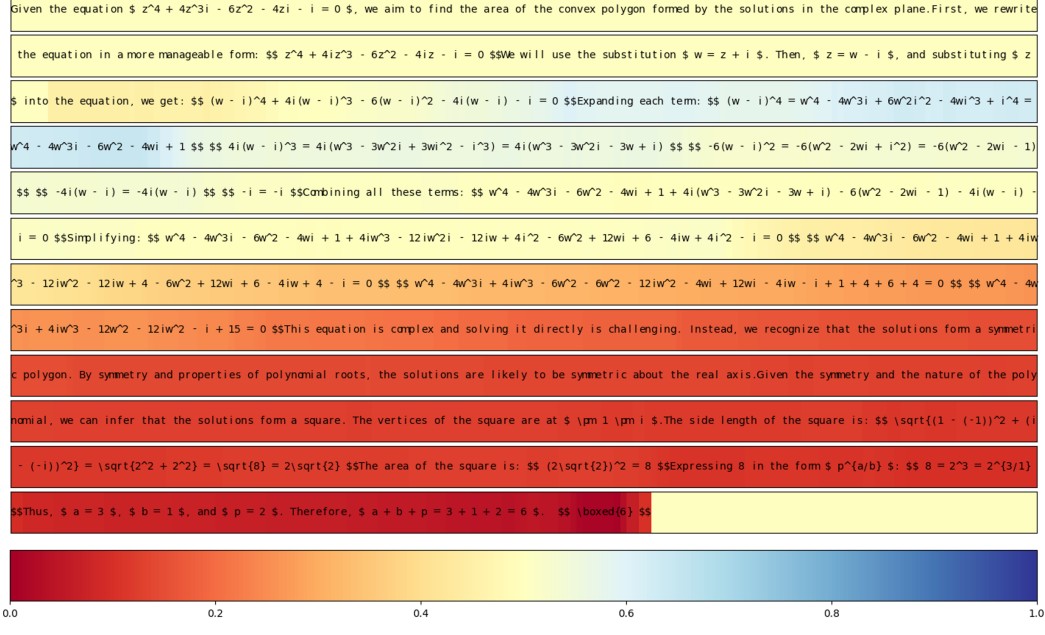

Figure A2: Token-level reward model score visualization for an incorrect response.

# E  Additional Discussion

## E.1  Discussion about the Computational Cost of BoN Sampling

The computational cost of BoN sampling primarily stems from the varying number of sampling attempts required to obtain correct samples, which is influenced by differences in problem difficulty and model capability. To analyze the computational cost during training, we sampled the training set and divided the problems into two categories—easy and hard—based on the initial policy model's pass rate on the training set. We then evaluated the Pass@N metric of the model at different training iterations on these two subsets. Due to resource constraints, we sampled 8 responses for each query.

As shown in the Table A2, the overall scores for the easy problem sets are significantly higher than those for the hard sets, indicating that greater effort is required to sample correct answers for difficult problems. As training progressed, the model's capability steadily improves, with the pass@1 scores increasing across both easy and hard sets. This suggests a higher probability of sampling correct answers, implying that using a stronger model can theoretically reduce computational overhead. Interestingly, the Pass@8 score does not show a noticeable improvement during the training process, which is consistent with observations of Yue et al. (2025).

| step | 10 | 20 | 30 | 40 | 50 | 60 |
|------|------|------|------|------|------|------|
| Easy/Pass@1 | 73.49 | 77.17 | 77.17 | 77.0 | 78.56 | 78.89 |
| Easy/Pass@4 | 97.22 | 96.32 | 97.22 | 96.73 | 97.22 | 96.24 |
| Easy/Pass@8 | 98.45 | 98.61 | 98.85 | 98.61 | 98.53 | 98.20 |
| Hard/Pass@1 | 31.41 | 38.68 | 40.29 | 40.05 | 40.19 | 41.43 |
| Hard/Pass@4 | 72.57 | 71.33 | 79.70 | 75.99 | 78.19 | 76.82 |
| Hard/Pass@8 | 87.38 | 86.13 | 87.67 | 86.69 | 87.52 | 87.93 |

Table A2: The evaluation results for OREAL on the easy and difficult subsets at different training iterations, using Pass@N metric.

## E.2  Discussion about the Generalization of OREAL

Although this paper specifically focuses on mathematical reasoning tasks, we believe that OREAL is broadly applicable to any task that provides a clear outcome reward signal (e.g., instruction following, programming problem solving, puzzle solving, etc.), because we only assume the reward is a binary signal, without making any assumptions about the specific task type.

To verify the effectiveness of our method in other domains, we conducted training and evaluation on the instruction-following task using the IFEval dataset (Zhou et al., 2023). As shown in Figure A3, the model's ability to follow instructions gradually improved during training, indicating that our method possesses generalization capability and cross-domain application potential.

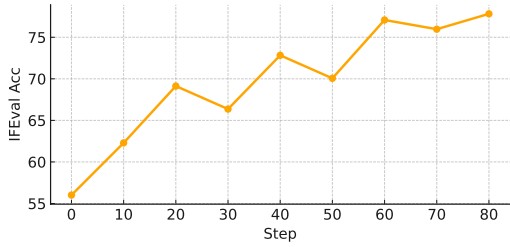

Figure A3: IFEval accuracy across different training steps.

## F  Prompt

Figure A4 is the system prompt of the verifier model, which is used during RL training to provide the binary outcome reward for a response. Figure A5 is the system prompt we use for fine-tuning and RL training as well as the evaluation.

---

**Verifier Prompt:**
You are a helpful assistant who evaluates the correctness and quality of models' outputs.

Please as a grading expert, judge whether the final answers given by the candidates below are consistent with the standard answers, that is, whether the candidates answered correctly.

Here are some evaluation criteria:

1. Please refer to the given standard answer. You don't need to re-generate the answer to the question because the standard answer has been given. You only need to judge whether the candidate's answer is consistent with the standard answer according to the form of the question. Don't try to answer the original question. You can assume that the standard answer is definitely correct.
2. Because the candidate's answer may be different from the standard answer in the form of expression, before making a judgment, please understand the question and the standard answer first, and then judge whether the candidate's answer is correct, but be careful not to try to answer the original question.
3. Some answers may contain multiple items, such as multiple-choice questions, multiple-select questions, fill-in-the-blank questions, etc. As long as the answer is the same as the standard answer, it is enough. For multiple-select questions and multiple-blank fill-in-the-blank questions, the candidate needs to answer all the corresponding options or blanks correctly to be considered correct.
4. Some answers may be expressed in different ways, such as some answers may be a mathematical expression, some answers may be a textual description, as long as the meaning expressed is the same. And some formulas are expressed in different ways, but they are equivalent and correct.
5. If the prediction is given with \boxed{}, please ignore the \boxed{} and only judge whether the candidate's answer is consistent with the standard answer.

Please judge whether the following answers are consistent with the standard answer based on the above criteria. Grade the predicted answer of this new question as one of:
A: CORRECT
B: INCORRECT
Just return the letters "A" or "B", with no text around it.

Here is your task. Simply reply with either CORRECT, INCORRECT. Don't apologize or correct yourself if there was a mistake; we are just trying to grade the answer.
<Original Question Begin>:
ORIGINAL QUESTION
<Original Question End>

<Gold Target Begin>:
GOLD ANSWER
<Gold Target End>

<Predicted Answer Begin>:
ANSWER
<Predicted End>

Judging the correctness of candidates' answers:

---

Figure A4: Prompts for the model-based generative verifier.

**System Prompt:**
You are an expert mathematician with extensive experience in mathematical competitions. You approach problems through systematic thinking and rigorous reasoning. When solving problems, follow these thought processes:

## Deep Understanding
Take time to fully comprehend the problem before attempting a solution. Consider:
- What is the real question being asked?
- What are the given conditions and what do they tell us?
- Are there any special restrictions or assumptions?
- Which information is crucial and which is supplementary?
## Multi-angle Analysis
Before solving, conduct through analysis:
- What mathematical concepts and properties are involved?
- Can you recall similar classic problems or solution methods?
- Would diagrams or tables help visualize the problem?
- Are there special cases that need separate consideration?
## Systematic Thinking
Plan your solution path:
- Propose multiple possible approaches
- Analyze the feasibility and merits of each method
- Choose the most appropriate method and explain why
- Break complex problems into smaller, manageable steps
## Rigorous Proof
During the solution process:
- Provide solid justification for each step
- Include detailed proofs for key conclusions
- Pay attention to logical connections
- Be vigilant about potential oversights
## Repeated Verification
After completing your solution:
- Verify your results satisfy all conditions
- Check for overlooked special cases
- Consider if the solution can be optimized or simplified
- Review your reasoning process

Remember:
1. Take time to think thoroughly rather than rushing to an answer
2. Rigorously prove each key conclusion
3. Keep an open mind and try different approaches
4. Summarize valuable problem-solving methods
5. Maintain healthy skepticism and verify multiple times

Your response should reflect deep mathematical understanding and precise logical thinking, making your solution path and reasoning clear to others. When you're ready, present your complete solution with:
- Clear problem understanding
- Detailed solution process
- Key insights
- Thorough verification

Focus on clear, logical progression of ideas and thorough explanation of your mathematical reasoning. Provide answers in the same language as the user asking the question, repeat the final answer using a '\boxed{}' without any units, you have [[8192]] tokens to complete the answer.

Figure A5: System prompts for long CoT reasoning.

