# OpenReview forum: "Exploring the Limit of Outcome Reward for Learning Mathematical Reasoning"
_colmweb.org/COLM/2025/Conference — COLM 2025_

### Official Review · Reviewer_XBDg · 2025-05-06

**Rating:** 6
**Confidence:** 1
**Ethics Flag:** 1

**Summary:**

This paper proposed a new reinforcement learning method to improve reasoning in LLMs.

**Questions To Authors:**

No

**Reasons To Accept:**

The  paper is well written
The proposed method appears to be novel
Experimental results are good.

I have limited understanding of this topic, so I’ll trust the experts’ judgment.

**Reasons To Reject:**

There is no comparison with Group Relative Policy Optimization (GRPO) in Deepseek.

---

> ### Author Response · Authors · 2025-06-02
>
> Thank you for your constructive comment. The following are our responses to each individual comment.
>
> ### **Response to Weakness about the missing comparison:**
>
> > There is no comparison with Group Relative Policy Optimization (GRPO) in Deepseek.
>
> We appreciate the comment and would like to clarify that we have indeed conducted both theoretical and experimental comparisons with GRPO.
>
> In the final paragraph of Section 2.3, we discuss the similarities between GRPO and our proposed method. Specifically, GRPO and our method both employ a similar reward shaping mechanism, but the key distinction lies in our approach, which applies this operation exclusively to negative samples, whereas GRPO applies it to all samples.
>
> Furthermore, Table 2 provides a comparative evaluation of these two strategies. The second row of Table 2 ("+reward shaping") represents the GRPO approach, while the last row showcases our proposed solution, demonstrating the superior performance of our method.

---

### Official Review · Reviewer_AWbc · 2025-05-11

**Rating:** 7
**Confidence:** 3
**Ethics Flag:** 1

**Summary:**

This paper introduces OREAL, a reinforcement learning (RL) framework for enhancing mathematical reasoning in large language models (LLMs), based solely on binary outcome feedback. The approach leverages three key insights: (1) behavior cloning from Best-of-N (BoN) positive trajectories is theoretically sufficient to recover a KL-regularized optimal policy, (2) reward shaping is essential for learning from negative trajectories due to BoN’s biased sampling distribution, and (3) a token-level reward model is used for token-wise credit assignment to alleviate sparse reward issues in long chain-of-thought (CoT) reasoning.

OREAL significantly improves performance on math reasoning benchmarks. A 7B model achieves 94.0 pass@1 on MATH-500—comparable to larger 32B models—and OREAL-32B achieves 95.0, setting a new state-of-the-art. The framework is shown to be effective across different model initializations, including both self-trained and distilled models.

**Questions To Authors:**

Minor comments are as below:
- While the paper consistently refers to sampling from a dataset $\mathcal{D}$, in Section 2.1 the distribution is denoted by ρ₀, which may cause confusion for readers. Clarifying this notation would improve readability.
- It would be useful to discuss computational cost of BoN sampling during training, especially for weaker base models or larger datasets.
- There is a minor typographical issue in Section 3.1, where the sentence refers to “Section ??”, likely due to an unresolved reference placeholder.

**Reasons To Accept:**

- **Well-motivated and theoretically grounded formulation of RL with binary rewards using BoN sampling**: The paper addresses a key challenge in mathematical reasoning: sparse, binary feedback. By focusing on outcome-level rewards, the authors design a framework well-suited to this setting. They show that behavior cloning from Best-of-N (BoN) sampled correct trajectories is sufficient to recover the KL-regularized optimal policy, while also demonstrating that negative trajectories, though underrepresented in BoN, can still contribute effectively through reward shaping. This provides a principled alternative to unstable REINFORCE-style methods and enables balanced learning from both successes and failures.

- **Clever use of distinct strategies for positive vs. negative samples**: The asymmetric treatment of correct and incorrect trajectories—behavior cloning for positives and reward shaping for negatives—is both novel and intuitive. It reflects the nature of reasoning tasks, where only some trajectories are optimal, and others still contain useful signals if handled properly. This design ensures balanced gradient updates despite BoN’s sampling bias.

- **Strong empirical results**: OREAL achieves new state-of-the-art performance on MATH-500 with a 7B model, even outperforming some 32B baselines, and shows solid results across other benchmarks like AIME and OlympiadBench.

**Reasons To Reject:**

- **Alternative framing via DPO**: A particularly interesting aspect of OREAL is the asymmetric learning strategy: behavior cloning on positive trajectories and reward shaping–based policy optimization on negative ones. This separation is both novel and intuitive. However, it also prompts a natural question: could a more unified preference-based approach, such as Direct Preference Optimization (DPO), be applied here by treating correct vs. incorrect outputs as preference pairs? If so, would it yield similar performance, possibly with simpler implementation or fewer hyperparameters? A comparative experiment with a DPO-style setup using the same BoN-derived data would provide valuable insights into the relative strengths of each formulation.

- **Limitation on initial policy strength**: The method seems to rely on having a reasonably good initial policy that can already generate some correct completions under BoN sampling. This assumption is implicitly reflected in the experimental setup, where initial policies are derived from either distilled or rejection-finetuned models. However, in real-world scenarios or low-resource domains, one may not have access to such strong initial models. It remains unclear how OREAL would perform when initialized from a significantly weaker model that produces mostly incorrect outputs. An empirical analysis of this case would help validate the framework's robustness and clarify its applicability in broader contexts beyond well-pretrained policies.

- **Dependency on the token-level reward model**: A potential limitation of the approach lies in its reliance on a token-level reward model trained solely from binary outcome labels. While this design avoids the need for step-level supervision and is computationally efficient, the paper does not provide a thorough analysis of the model’s stability or accuracy, especially under distribution shifts or varying reasoning patterns. Since the token-level weights directly influence the policy optimization, any misestimation could affect learning dynamics, and further investigation into the robustness and generalizability of this component would strengthen the overall claim.

---

> ### Author Response · Authors · 2025-06-02
> **Response by Authors for Weakness1-2**
>
> Thank you for your constructive comment. The following are our responses to each individual comment.
>
> ### **Response to Weakness1 about the alternative framing:**
>
> > Alternative framing via DPO: A particularly interesting aspect of OREAL is the asymmetric learning strategy: behavior cloning on positive trajectories and reward shaping–based policy optimization on negative ones. This separation is both novel and intuitive. However, it also prompts a natural question: could a more unified preference-based approach, such as Direct Preference Optimization (DPO), be applied here by treating correct vs. incorrect outputs as preference pairs? If so, would it yield similar performance, possibly with simpler implementation or fewer hyperparameters? A comparative experiment with a DPO-style setup using the same BoN-derived data would provide valuable insights into the relative strengths of each formulation.
>
> Yes, after distinguishing between positive and negative samples, DPO can be used for training. However, DPO has a potential flaw: it theoretically leads to a decrease in the generation probability of positive examples, even while increasing their relative generation probability compared to negative examples [1]. In contrast, our approach performs Behavior Cloning for positive samples, avoids this particular issue, which is precisely why we chose not to employ DPO.
>
> [1] Pal A, Karkhanis D, Dooley S, et al. Smaug: Fixing failure modes of preference optimisation with dpo-positive[J]. arXiv preprint arXiv:2402.13228, 2024.
>
> ### **Response to Weakness2 about the limitation on initial policy strength:**
>
> > Limitation on initial policy strength: The method seems to rely on having a reasonably good initial policy that can already generate some correct completions under BoN sampling. This assumption is implicitly reflected in the experimental setup, where initial policies are derived from either distilled or rejection-finetuned models. However, in real-world scenarios or low-resource domains, one may not have access to such strong initial models. It remains unclear how OREAL would perform when initialized from a significantly weaker model that produces mostly incorrect outputs. An empirical analysis of this case would help validate the framework's robustness and clarify its applicability in broader contexts beyond well-pretrained policies.
>
> In Appendix C, we discuss the performance of our method across different initial policy models. As shown in Table A1, the performance of the model after RL is strongly correlated with the capabilities of the initial policy model itself. The stronger the initial policy model, the higher the performance that RL can deliver, indicating the importance of policy initialization.
>
> Nevertheless, it is noteworthy that our method consistently makes substantial performance improvements, regardless of the initial policy model's strength, which demonstrates the robustness of our approach.
>
> We acknowledge that for particularly challenging data (e.g., AIME-level problems), if the model fails to sample correct answers, the training effectiveness may indeed be compromised. However, this is an inevitable problem that all on-policy reinforcement learning methods have to face.

---

> > ### Comment · Reviewer_AWbc · 2025-06-08
> >
> > Thank you for the detailed responses. I confirm that the authors have adequately addressed most of the initial concerns and questions.

---

> > > ### Author Response · Authors · 2025-06-08
> > >
> > > Thank you for your response. We are happy that our discussions are convincing. We will update the manuscript based on your comments in the next version.

---

> ### Author Response · Authors · 2025-06-02
> **Response by Authors for Weakness3 and Question1-3**
>
> Thank you for your constructive comment. The following are our responses to each individual comment.
>
> ### **Response to Weakness3 about the dependency on the token-level reward model:**
>
> > Dependency on the token-level reward model: A potential limitation of the approach lies in its reliance on a token-level reward model trained solely from binary outcome labels. While this design avoids the need for step-level supervision and is computationally efficient, the paper does not provide a thorough analysis of the model’s stability or accuracy, especially under distribution shifts or varying reasoning patterns. Since the token-level weights directly influence the policy optimization, any misestimation could affect learning dynamics, and further investigation into the robustness and generalizability of this component would strengthen the overall claim.
>
> We acknowledge that our method relies on the token-level reward model training on binary outcome labels, but we believe it is a low-cost yet reliable solution.
>
> **Regarding the potential issue of distribution shift,** it needs to be clarified that in our scheme, the token-level reward model is trained synchronously with the policy model using the same training data. This ensures that the data it uses for training and the data it needs to estimate within a single iteration are co-distributed, thereby mitigating the risk of misestimation due to distribution shift.
>
> **About the reliability of this reward model,** we provide a visualization of its estimated values for both positive and negative samples in Appendix D2. For correct responses, the overall reward scores are high, especially at the end, although there are a few lower sections in the middle. For incorrect responses, the distribution of rewards is reversed, and the closer to the end, the lower the rewards. We believe the distinct distribution of scores for positive and negative samples offers evidence for the model's stability and accuracy.
>
> ### **Response to Question1 about the unclear presentation:**
>
> > While the paper consistently refers to sampling from a dataset D, in Section 2.1, the distribution is denoted by ρ₀, which may cause confusion for readers. Clarifying this notation would improve readability.
>
> Thanks for your suggestion. We will modify these confusing variable names in the next version to improve readability.
>
> ### **Response to Question2 about the insufficient disscussion:**
>
> > It would be useful to discuss computational cost of BoN sampling during training, especially for weaker base models or larger datasets.
>
> Thanks for your suggestion. The computational cost of BoN sampling primarily stems from the varying number of sampling attempts required to obtain correct samples, which is influenced by differences in problem difficulty and model capability. To analyze the computational cost during training, we sampled the training set and divided the problems into two categories—easy and hard—based on the initial policy model's pass rate on the training set. We then evaluated the Pass@N metric of the model at different training iterations on these two subsets. Due to resource constraints, we sampled 8 responses for each query.
>
> As shown in the table below, the overall scores for the easy problem sets are significantly higher than those for the hard sets, indicating that greater effort is required to sample correct answers for difficult problems. As training progressed, the model's capability steadily improves, with the pass@1 scores increasing across both easy and hard sets. This suggests a higher probability of sampling correct answers, implying that using a stronger model can theoretically reduce computational overhead. Interestingly, the pass@8 score does not show a noticeable improvement during the training process, which is consistent with observations of Yue et al.[2].
>
> | step       | 10     | 20     | 30     | 40     | 50     | 60     |
> |------------|--------|--------|--------|--------|--------|--------|
> | easy/pass@1 | 73.49  | 77.17  | 77.17  | 77.0   | 78.56  | 78.89  |
> | easy/pass@4 | 97.22  | 96.32  | 97.22  | 96.73  | 97.22  | 96.24  |
> | easy/pass@8 | 98.45  | 98.61  | 98.85  | 98.61  | 98.53  | 98.20  |
> | hard/pass@1 | 31.41  | 38.68  | 40.29  | 40.05  | 40.19  | 41.43  |
> | hard/pass@4 | 72.57  | 71.33  | 79.70  | 75.99  | 78.19  | 76.82  |
> | hard/pass@8 | 87.38  | 86.13  | 87.67  | 86.69  | 87.52  | 87.93  |
>
> We will add a corresponding discussion and evaluation results in the next version.
>
> [2] Yue Y, Chen Z, Lu R, et al. Does reinforcement learning really incentivize reasoning capacity in llms beyond the base model?[J]. arXiv preprint arXiv:2504.13837, 2025.
>
> ### **Response to Question3 about the typos:**
>
> > There is a minor typographical issue in Section 3.1, where the sentence refers to “Section ??”, likely due to an unresolved reference placeholder.
>
> Thanks for your reminder. We will fix the typos in the next version.

---

### Official Review · Reviewer_qoai · 2025-05-12

**Rating:** 7
**Confidence:** 2
**Ethics Flag:** 1

**Summary:**

This paper proposes OREAL, a reinforcement learning framework that utilizes binary outcome rewards for mathematical reasoning tasks, and explores its theoretical background and practical effectiveness. OREAL is founded on three pillars: imitation of correct solution trajectories, reward design for negative examples, and token-level importance weighting. The method achieves higher performance in mathematical reasoning models compared to conventional approaches. The paper demonstrates that OREAL significantly outperforms existing methods on 7B and 32B scale language models, and notably, shows that the 7B model reaches state-of-the-art performance using reinforcement learning alone for the first time.

**Reasons To Accept:**

- Clear theoretical foundation and mathematical interpretation
- Comprehensive experiments with detailed ablation studies
- Implementation of credit assignment through token-level rewards
- Confirmed superiority through quantitative comparisons with competitive baselines

**Reasons To Reject:**

Insufficient explanation of algorithm generality and cross-domain applications

---

> ### Author Response · Authors · 2025-06-02
>
> Thank you for your constructive comment. The following are our responses to each individual comment.
>
> ### **Response to Weakness about the insufficient explanation:**
>
> > Insufficient explanation of algorithm generality and cross-domain applications
>
> We acknowledge that this paper has limitations in discussing the generalization and cross-domain applicability of our algorithm. As indicated by the title, this paper specifically focuses on mathematical reasoning tasks. However, we believe that our method is broadly applicable to any task that provides a clear outcome reward signal (e.g., instruction following, programming problem solving, puzzle solving, etc.), because we only assume the reward is a binary signal, without making any assumptions about the specific task type.
>
> To verify the effectiveness of our method in other domains, we conducted training and evaluation on the instruction-following task using the IFEval dataset [1]. As shown in the table below, the model's ability to follow instructions gradually improved during training, indicating that our method possesses generalization capability and cross-domain application potential.
>
> | step       | 0     | 10    | 20    | 30    | 40    | 50    | 60    | 70    | 80    |
> |------------|-------|-------|-------|-------|-------|-------|-------|-------|-------|
> | IFEval acc   | 56.01 | 62.29 | 69.13 | 66.36 | 72.83 | 70.06 | 77.08 | 75.97 | 77.82 |
>
> We will add a corresponding discussion and evaluation results in the next version.
>
>
> [1] Zhou J, Lu T, Mishra S, et al. Instruction-following evaluation for large language models[J]. arXiv preprint arXiv:2311.07911, 2023.

---

> ### Comment · Reviewer_qoai · 2025-06-03
>
> I appreciate the follow-up experiment. My concern has been resolved.

---

> > ### Author Response · Authors · 2025-06-03
> >
> > Thank you for your response. We are happy that our discussions are convincing. We will update the manuscript based on your comments in the next version.

---

### Official Review · Reviewer_cCHz · 2025-05-21

**Rating:** 6
**Confidence:** 3
**Ethics Flag:** 1

**Summary:**

This paper studies the design choices of RL algorithms for teaching the LLM mathematical reasoning. The resulting algorithm Outcome
REwArd-based reinforcement Learning  (OREAL) achieves sota performance on several benchmarks.

**Questions To Authors:**

For the REINFORCE baseline in Table 2, is it based on outcome-reward or process-reward?

**Reasons To Accept:**

1. The paper proposes a practical algorithm that contains various components, some of which are novel.
2. The experimental results are strong, surpass the baselines by a large margin.
3. The ablation is well conducted, validating the effect of each component, including reward shaping, behavior cloning, importance sampling, and skill-based enhancement.
4. Most experiment setups are clear.

**Reasons To Reject:**

1. The method is a combination of several components with various tricks being applied, making it hard to distinguish the algorithmic novelty.
2.  Some descriptions of the method are confusing. For instance, what does importance sampling refer to in the method, e.g., which expectation is IS estimating? What is reward shaping referring to? Is it different from traditional reward shaping that uses value potentials?
3. The name Outcome Reward RL is somewhat misleading since the algorithm uses reward models to obtain dense process rewards, even though the reward model is trained using the outcome reward.

---

> ### Author Response · Authors · 2025-06-02
>
> Thank you for your constructive comment. The following are our responses to each individual comment.
>
> ### **Response to Weakness1 about the algorithmic novelty:**
>
> > The method is a combination of several components with various tricks being applied, making it hard to distinguish the algorithmic novelty.
>
> Our method is not merely a combination of several components with various tricks being applied, it offers a theoretically grounded reinforcement learning algorithm for mathematical reasoning tasks. Specifically, our main contribution lies in demonstrating, through rigorous theoretical analysis and extensive experimentation, the following:
>
> - **For Positive Samples:** In binary feedback environments, optimal policies can be effectively learned through behavior cloning on best-of-N sampled trajectories.
>
> - **For Negative Samples:** Reward reshaping is essential to maintain consistency in the policy optimization objective.
>
> - **For Long Sequences:** Different segments of a long sequence contribute disparately to the final outcome, necessitating a more fine-grained credit assignment function. This function can be effectively learned from the outcome reward itself.
>
> ### **Response to Weakness2 about the unclear presentation:**
>
> > Some descriptions of the method are confusing. For instance, what does importance sampling refer to in the method, e.g., which expectation is IS estimating? What is reward shaping referring to? Is it different from traditional reward shaping that uses value potentials?
>
> **For the unclear presentation about "Importance Sampling".** We agree that our use of the term importance sampling may be misleading. In our method, we do not perform importance sampling in the classical sense that policy updates are constrained by the difference between two distributions, such as using log-likelihood ratios between the new and old policies. Instead, we use a token-level reward model which trained on historical policy data to assign scores reflecting each token’s estimated contribution to the final reward. These scores perform fine-grained credit assignments under long reasoning sequences and are used as heuristic importance weights to prioritize the learning of high-contributing tokens and downweight or ignore low-contributing tokens. This is akin to importance-based selection or re-weighting, rather than the traditional meaning of importance sampling.
>
> To improve clarity, we will revise the terminology to 'importance-based token selection' or 'fine-grained token re-weighting' rather than importance sampling in the final version.
>
> **For the unclear presentation about "Reward Shaping".**  As shown in Equation 2, since our positive and negative samples are derived from the results of BoN sampling, their reward distribution is inconsistent with that of uniformly sampling from the original distribution. Generally, this will overestimate the expected reward of the current policy on the corresponding input (high-reward responses are assigned a higher probability), which may affect downstream reasoning performance (as the optimization objective is not consistent with the pass@1 performance). Therefore, we introduce a coefficient independent of the model parameters to shape the negative sample rewards, theoretically ensuring that our optimization objective function exactly remains the original expected reward.
>
> Regarding "traditional reward shaping that uses value potentials", which is typically associated with advantage estimation in traditional RL algorithms, we further discuss it in Section 2.4 and explain why we use the estimated advantage function for loss weighting instead of directly using it to calculate the optimization objective.
>
> ### **Response to Weakness3 about the misleading title:**
>
> > The name Outcome Reward RL is somewhat misleading since the algorithm uses reward models to obtain dense process rewards, even though the reward model is trained using the outcome reward.
>
> We name our work "Outcome Reward RL" because our training objective is to maximize a binary outcome reward. In this paper, we observe that for long-cot mathematical reasoning tasks, different parts of the sequence contribute differently to the final outcome, while relying solely on binary outcome rewards leads to the problem of reward sparsity. To address this, we introduce a token-level reward model to perform token-wise credit assignments during training. It is important to clarify that this token-level reward model is only used in the computation of importance sampling and does not serve as our final optimization objective, whose role is to assist in more effectively maximizing the overall outcome reward. Therefore, we believe that our method represents a reinforcement learning approach grounded in outcome-based rewards.
>
> ### **Response to Question1 about the details of Table2:**
>
> > For the REINFORCE baseline in Table 2, is it based on outcome-reward or process-reward?
>
> It is based on outcome-reward.

---

> > ### Comment · Reviewer_cCHz · 2025-06-02
> > **Response to authors**
> >
> > Thank you for your responses, which address most of my concerns. I hope the authors could add the discussions, especially the clarification on the unclear terms and the use of process rewards, to a later version of the manuscript.

---

> > > ### Author Response · Authors · 2025-06-02
> > >
> > > Thank you for your response. We are happy that our discussions are convincing. We will update the manuscript based on your comments in the next version.

---

### Decision · Program_Chairs · 2025-07-08

**Decision:**

Accept

**Comment:**

This paper proposes OREAL, a RL framework designed to improve mathematical reasoning in LLMs using only binary outcome rewards. It introduces a theoretically grounded approach that combines behavior cloning from best-of-N samples, reward shaping for negative examples, and token-level credit assignment. The method achieves strong empirical results, notably enabling a 7B model to match the performance of much larger models on MATH-500.
Reviewers all recommend accepting the paper, with positive feedback highlighting the importance of the problem, the clarity of the theoretical formulation, and the strength of the experimental results. Several reviewers appreciated the novel combination of techniques and the insight into RL challenges in long-chain reasoning tasks. The authors actively engaged in the discussion phase, well addressing reviewer concerns and adding additional evaluations on generalization and comparisons with existing methods (e.g., GRPO, DPO). Overall, this paper makes a meaningful contribution to RL-based reasoning with LLMs and I recommend acceptance.